# High Prolactin Concentration Induces Ovarian Granulosa Cell Oxidative Stress, Leading to Apoptosis Mediated by *L-PRLR* and *S-PRLR*

**DOI:** 10.3390/ijms241914407

**Published:** 2023-09-22

**Authors:** Ruochen Yang, Chunhui Duan, Shuo Zhang, Yunxia Guo, Xinyu Shan, Meijing Chen, Sicong Yue, Yingjie Zhang, Yueqin Liu

**Affiliations:** 1College of Animal Science and Technology, Hebei Agricultural University, Baoding 071000, China; yangruochen1110@126.com (R.Y.); duanchh211@126.com (C.D.); 13932229451@163.com (X.S.); chenmeijing815@126.com (M.C.); yueyueyuexiaojuan@163.com (S.Y.); 2College of Animal Science and Technology, China Agricultural University, Beijing 100089, China; 17835424912@163.com; 3College of Life Sciences, Hebei Agricultural University, Baoding 071000, China; gyx310@163.com

**Keywords:** ovine ovarian GCs, apoptosis, oxidative stress, autophagy, mitochondria, ROS, *PRLR*s

## Abstract

High prolactin (PRL) concentration has been shown to induce the apoptosis of ovine ovarian granulosa cells (GCs), but the underlying mechanisms are unclear. This study aimed to investigate the mechanism of apoptosis induced by high PRL concentration in GCs. Trial 1: The optimal concentration of glutathion was determined according to the detected cell proliferation. The results showed that the optimal glutathione concentration was 5 μmol/mL. Trial 2: 500 ng/mL PRL was chosen as the high PRL concentration. The GCs were treated with 0 ng/mL PRL (C group), 500 ng/mL PRL (P group) or 500 ng/mL PRL, and 5 μmol/mL glutathione (P-GSH group). The results indicated that the mitochondrial respiratory chain complex (MRCC) I–V, ATP production, total antioxidant capacity (T-AOC), superoxide dismutase (SOD), and thioredoxin peroxidase (TPx) in the C group were higher than those in the P group (*p* < 0.05), while they were lower than those in the P-GSH group (*p* < 0.05). Compared to the C group, the P group exhibited elevated levels of reactive oxygen species (ROS) and apoptosis (*p* < 0.05) and increased expression of ATG7 and ATG5 (*p* < 0.05). However, MRCC I–V, ATP, SOD, A-TOC, TPx, ROS, and apoptosis were decreased after the addition of glutathione (*p* < 0.05). The knockdown of either *L-PRLR* or *S-PRLR* in P group GCs resulted in a significant reduction (*p* < 0.05) in MRCC I–V, ATP, T-AOC, SOD and TPx, while the overexpression of either receptor showed an opposite trend (*p* < 0.05). Our findings suggest that high PRL concentrations induce apoptotic cell death in ovine ovarian GCs by downregulating *L-PRLR* and *S-PRLR*, activating oxidative stress and autophagic pathways.

## 1. Introduction

Prolactin (PRL), as the most versatile pituitary hormone, holds a crucial position in the biological action and regulation of numerous physiological processes [1]. In females, it stimulates steroidogenesis [2], particularly in the corpus luteum, and stimulates the growth of ovarian follicles [3]. The apoptosis of GCs is a physiological phenomenon of follicular development that can maintain the normal number of mature follicles in the ovary through follicular atresia [4]. However, the excessive apoptosis of GCs may have a negative effect on the cell connection between the GCs themselves and between GCs and the oocyte, leading to female reproductive diseases such as premature ovarian failure [5], hyperprolactinemia [6] and polycystic ovarian syndrome (PCOS) [7]. The previous study showed that 4 ng/mL PRL promotes proliferation of ovine GCs. However, some research [8,9] has suggested that apoptosis of the ovarian GCs depends on the increased amount of PRL administered. Zamani et al. found that chlorpromazine-induced hyperprolactinemia caused a significant decrease in the number of follicles and atresia of the normal corpora lutea [10]. In both hyperprolactinemia and PCOS [11], elevated levels of PRL have been observed, indicating a possible association between aberrant apoptosis of GCs and hormonal imbalances. Moreover, the expression of the PRL receptor (*PRLR*) was detected on the surface of GCs in female ovaries, suggesting that PRL can bind to *PRLR* on GCs and affect the function of GCs directly [12]. In the chronic mild stress mouse model of depression, a previous study showed that apoptosis of CA3 hippocampal neurons was associated with reduced expression of *PRLR* [13]. Moreover, *PRLR* knockdown has been shown to increase apoptosis in a mouse pancreatic beta cell line [14]. Ruminants have two types of *PRLR*, long *PRLR* (*L-PRLR*) and short *PRLR* (*S-PRLR*) [15], but the specific relationship between *L-PRLR*, *S-PRLR* and GC apoptosis has not been reported.

Oxidative stress is defined as a consequence of disruption in the equilibrium of intra-cellular reactive oxygen species (ROS) generation, including hydrogen peroxide (H_2_O_2_), hydroxyl radicals (−OH) and superoxide anion (O_2_^–^), and their scavenger by antioxidants, has been identified as one of the key factors of ovarian dysfunction [16,17] and female infertility [18]. The occurrence of oxidative stress resulted in aberrant cellular proliferation, functional abnormalities and apoptosis of bovine ovarian GCs [19], while the fertilisation rate decreased and DNA damage increased in cases of oxidative stress detected in the follicular fluid of human GCs [20]. Recent studies have shown that autophagy defects in human GCs were observed in PCOS ovaries at different stages of follicles [21], accompanied by an increased concentration of PRL [22]. Autophagy played a key role in proliferation and ovulation in GCs [23], suggesting that high PRL concentration could regulate cellular apoptosis by affecting cellular autophagy. Several processes, including oxidative stress, autophagy and apoptosis, are related to the functions of ovaries. Studies have found that oxidative stress could induce apoptosis by activating ROS [24] and mitochondrial damage [25]. Autophagy and apoptosis were also closely related [26]. Numerous autophagy molecules participating in the process of autophagy may be responsible for the induction of cell apoptosis, thus aggravating cell damage [27]. However, whether high PRL concentration regulates apoptosis in ovine ovarian GCs through oxidative stress and autophagy requires further investigation.

Our previous study revealed the high PRL concentration induced GCs oxidative stress and autophagy [28]. Thus, the present study hypothesised that high PRL concentration induced apoptosis of ovine ovarian GCs may be through oxidative stress and autophagy. We employed CRISPR/Cas9 and the overexpression technology with an integrated bioinformatics approach to explore the mechanisms of high prolactin concentration-induced GCs apoptosis. It will provide a basis for future applications of PRL to follicle development and the reproduction in ruminant.

## 2. Results

### 2.1. Cell Proliferation and Expression of Apoptosis-Related Genes

Cell viability was evaluated in different concentrations of the glutathione. The results indicated that with an increase in glutathione concentration, cell viability increased firstly, then it decreased (Figure 1A). The level of cell viability in the 0 μmol/mL, 1 μmol/mL, 2 μmol/mL, and 10 μmol/mL groups were lower than that in the 5 μmol/mL group (*p* < 0.05).

The expression of apoptosis-related genes (*Bcl-2*, *Bax* and *Caspase3*) in the C (Control group: GCs with 0 ng/mL glutathione) and T (GCs with 5μmol/mL glutathione) groups are shown in Figure 1B. The expression level of *Bcl-2* and *Caspase3* was higher in the T group than that in the C group (*p* < 0.05). The expression of *Bax* was lower in the T group than that in C group (*p* < 0.01), indicating that cell apoptosis can be inhibited by administrating 5 μmol/mL glutathione.

### 2.2. Identification of Knockdown and Overexpression of L-PRLR and S-PRLR

Gene editing technology was used to knock down and overexpress *L-PRLR* and *S-PRLR*, respectively, to explore whether high PRL concentration regulates oxidative stress through *L-PRLR* and *S-PRLR*.

We determine whether the overexpression is successful by fluorescence observation and qPCR detection because the overexpressed plasmid pGWLV10-new carries the GFP fluorescent green label. The cell fluorescence of each group after the overexpression of *L-PRLR* and *S-PRLR* was shown in Figure 2A–C. RT-qPCR was performed for the infected groups, and the results are shown in Figure 2G,H. The expression of *L-PRLR* in P-10-L group was significantly higher than that in P and P-10 groups (*p* < 0.01), The expression of *S-PRLR* in P-10-S group was higher than that in P and P-10 groups (*p* < 0.05). The expression of *L-PRLR* and *S-PRLR* in both P and P-10 groups have no significant difference, indicating that the overexpression was successful. Both *L-PRLR* and *S-PRLR* had on difference between P and P-10 groups, indicating that the pGWLV10-new vector enveloped by lentivirus had no significant effect on the expression of *L-PRLR* and *S-PRLR*.

The cell morphology of each group after the knockdown of *L-PRLR* and *S-PRLR* was shown in Figure 2D–F. RT-qPCR was performed for the infected groups, and the results are shown in Figure 2G,H. After infections, *L-PRLR* was downregulated in the P-sg-L group compared to the P and P-sg groups (*p* < 0.05), while in the P-sg-S group, the expression of *S-PRLR* was lower than that in the P and P-sg groups (*p* < 0.05). The expression of *L-PRLR* and *S-PRLR* in both P and P-sg groups had no significant difference, indicating that the knockdowns were successful. In addition, the expression of *L-PRLR* and *S-PRLR* was no different between the P and P-sg groups, suggesting that the lenticrispr vector2 enveloped by lentivirus had no significant effect on the expression of *L-PRLR* and *S-PRLR*.

The results of the expression of *L-PRLR* and *S-PRLR* in C, P and P-GSH groups were shown in Figure 2I. Compared to the C group, the expression of *L-PRLR* and *S-PRLR* was reduced in the P group (*p* < 0.05). However, the expression of *L-PRLR* and *S-PRLR* was higher in the P-GSH group than that in the P group (*p* < 0.01).

### 2.3. Mitochondrial Complex Activity Analysis

Figure 3 depicts the structure of the mitochondrial complexes and the generative pro cess ATP, and the activity of mitochondrial complexes (I, II, III, IV and V), as well as the content of ATP, are shown in Figure 4 and Figure 5. After treatment with 500 ng/mL PRL, the activity of MRCC I, II, III, V, and the ATP content were significantly reduced compared to the C group (*p* < 0.05); however, the addition of the antioxidant glutathione resulted in a significant increase in these activity levels and ATP content (*p* < 0.05), indicating that high PRL concentration inhibited the production of ATP through oxidative stress. The activity of MRCC I, II, and III and the ATP content were repressed in P-sg-L and P-sg-SL groups (*p* < 0.05) compared with P group, while they were elevated in P-10-L and P-10-S groups (*p* < 0.05) compared to that in the P group. The activity of MRCC IV and V was lower in P-sg-L and P-sg-SL groups (*p* < 0.05) compared to the P group, while being significantly higher in the P-10-L group (*p* < 0.05) than that in the P group.

### 2.4. Oxidative Stress Parameters Analysis

The forward scatter area and FSC-A and FITC-A of ROS of the negative and positive control are shown in Figure 6A,B. The results of the ROS positive rate is shown in Figure 6C,D, and the results of the T-AOC, SOD and TPx in each group are shown in Figure 7. The level of ROS was significantly increased and the levels of T-AOC, SOD and TPx were decreased in P group as compared to the C group (*p* < 0.01), while the levels of ROS, T-AOC, SOD and TPx (*p* < 0.05) showed the opposite trend after the addition of antioxidant glutathione, indicating that high PRL concentration lead to oxidative stress.

The ROS level was higher in P-sg-L and P-sg-SL groups (*p* < 0.05) than in P group, while it was lower in the P-10-L and P-10-S groups (*p* < 0.01) compared with the P group. On the other hand, the ROS level was significantly higher in the P-sg-SL group (*p* < 0.05), when compared to the P-sg-L group. After infections, the levels of T-AOC, SOD, and TPx declined in the P-sg-L and P-sg-SL groups (*p* < 0.05) compared to the P group, while they were elevated in the P-10-L and P-10-S groups (*p* < 0.01) compared to the P group. Furthermore, the T-AOC, SOD, and TPx in the P-sg-L group were significantly lower (*p* < 0.01) than in the P-sg-SL group. These results suggested that a high PRL concentration induced oxidative stress through *L-PRLR* and *S-PRLR*.

### 2.5. Apoptosis and Cell Viability Assay

The late withered, early withered, living, and dead cells in each group were distinguished by the detection results (Figure 8A–G). As for the early apoptotic cells rate, late apoptotic cells rate, and total apoptotic rate of GCs (Table 1), they were upregulated in P group compared to the C group (*p* < 0.05), while they were downregulated (*p* < 0.05) after the addition of antioxidant glutathione, indicating that high PRL concentrations lead to apoptosis through oxidative stress. The early apoptotic cells rate, late apoptotic cells rate, and total apoptotic rate were elevated in the P-sg-L and P-sg-SL groups (*p* < 0.05) compared to the P group, while they were repressed in P-10-L and P-10-S groups (*p* < 0.05) than that in P group. And the early apoptotic cells rate, late apoptotic cells rate, and total apoptotic rate were higher in P-sg-SL group than P-sg-L group (*p* < 0.05). These data suggested that high PRL concentration could promote the cell apoptosis rate by downregulating the expression of *L-PRLR* and *S-PRLR*.

The results of the activity of GCs (Figure 8H,I) were consistent with the trend of apoptosis. The activity of GCs was significantly decreased in the P group as compared to the C group (*p* < 0.01), while the activity of GCs (*p* < 0.01) showed the opposite trend after the addition of antioxidant glutathione. The activity of GCs was lower in the P-sg-L and P-sg-SL groups (*p* < 0.05) than in the P group, but it was higher in the P-10-L and P-10-S groups (*p* < 0.01) compared with the P group. On the other hand, the activity of GCs was significantly lower in the P-sg-SL group (*p* < 0.05) when compared to the P-sg-L group.

### 2.6. Autophagy Associated Proteins Analysis

The results of the autophagy associated proteins (ATG7 and ATG5) expression are shown in Figure 9. The protein expressions of ATG7 and ATG5 were increased in the P group (*p* < 0.05) compared to the C group. Furthermore, the expression of ATG7 and ATG5 was decreased in P-GSH group (*p* < 0.05) in relation to the P group. These dates suggested that high PRL concentration accelerated the autophagy of GCs, which was alleviated by the addition of antioxidant glutathione.

After infections, the protein expressions of ATG7 and ATG5 were shown to be increased in the P-sg-L and P-sg-SL groups (*p* < 0.05) compared to the P group, but were downregulated in the P-10-L and P-10-S groups (*p* < 0.01) compared to the P group. In addition, the expression of ATG7 and ATG5 in the P-sg-SL group was higher than that in P-sg-L group. The results indicated that a high PRL concentration induced the autophagy of GCs through *L-PRLR* and *S-PRLR*.

## 3. Discussion

### 3.1. High PRL Concentrations Caused Mitochondrial Damage in Ovine Ovarian GCs and Excessive ROS

Mitochondria generate ATP, which is required for cell growth and development. Mitochondrial oxidative phosphorylation facilitates the production of ATP through a series of coupled reactions involving mitochondrial respiratory chain complexes I–V [29]. At the same time, it is accompanied by the generation of ROSs [30]. A previous study demonstrated that L-817,818 ameliorated the function of MRCC in a rat experimental glaucoma model through decreasing the levels of ROSs [31]. ROSs are eliminated by antioxidants as a by-product of oxidative metabolism under normal circumstances. However, mitochondrial dysfunction promotes an increase in ROSs, and excessive ROSs damage genes, proteins, and lipids in mitochondria and cells, leading to further impairment of the mitochondrial respiratory chain and oxidative stress [32]. Previous studies have shown that serum high PRL concentration could contribute to infertility by inducing oxidative damage [33,34]. SOD, T-AOC, and TPx are important indicators that reflect the oxidative stress status of the organism. SOD is a key component of the antioxidant enzymatic defence system; the level of T-AOC represents the overall cellular endogenous antioxidative capabilities; and TPx activity plays an important role in protecting against oxidative stress [35]. The activity of SOD, T-AOC, TPx, and MRCCI–V decreased when oxidative stress occurred in the body [31,36]. In this study, the SOD, T-AOC, TPx, and MRCC I, II, III, and V activities were significantly decreased in ovine ovarian GCs supplemented with 500 ng/mL PRL. More importantly, the ROS content in GCs was significantly increased after the addition of 500 ng/mL PRL, indicating that a high PRL concentration could cause oxidative stress in GCs through ROS and mitochondrial damage, which consistent with the results of the previous study. Khalaf et al. showed that the serum PRL concentration in HAL-treated rats increased significantly, while the activities of MRCC I, MRCC III, SOD, CAT, and TPx were significantly decreased. Histopathological studies of HAL-treated ovaries have shown features suggestive of hyperprolactinemia and oxidative stress [37]. A previous study showed that ROS homeostasis is one of the key factors in maintaining ovarian follicle development and that oxidative stress occurs when the balance between ROS and antioxidants is disrupted, leading to the apoptosis of GCs and oocytes [38]. In this study, the apoptosis rate of GCs was significantly increased after the addition of 500 ng/mL PRL, while it significantly decreased after the addition of the antioxidant glutathione, suggesting the apoptosis of high PRL concentration-induced apoptosis through oxidative stress, similar to previous studies [38].

### 3.2. High PRL Concentrations Induced Ovine Ovarian GC Apoptosis by Promoting Oxidative Stress and Autophagy

The regulation of autophagy has been identified to play a role in significant cell death events occurring during the ovarian cycle [39]. Autophagy is a cellular catabolic mechanism that removes and degrades damaged organelles and macromolecular substances to protect cells from stress conditions and maintain cell homeostasis [40]. ATG5 and ATG7 are considered key proteins for the induction of autophagy. Autophagy requires the activity of two autophagy-specific ubiquitin-like conjugated systems, the ATG12–5 (ATG16L complex) and the LC3 (ATG8) conjugation system, both of which require ATG7. In this study, the protein contents of ATG5 and ATG7 were significantly increased after the addition of 500ng/mL PRL, which was consistent with the changing trend of the GC apoptosis rate. Therefore, we believe that a high PRL concentration induced excessive autophagy of GCs and accelerated apoptosis. Interestingly, the expression levels of autophagy-related proteins ATG5 and ATG7 were significantly decreased after the addition of antioxidants, which could be due to the interaction between autophagy and cellular oxidative stress. A previous study found that the increased levels of PRL, P_4_, and E_2_ in the GCs of egg goose follicles were accompanied by increased levels of autophagy, indicating that PRL is related to the autophagy of GCs in follicles [41]. In this experiment, the level of autophagy was significantly increased in the high PRL concentration group, consistent with the results of previous studies. Nevertheless, autophagy can also induce cell death through its interaction with apoptosis. Researchers have different opinions about the relationship between autophagy and apoptosis. In general, autophagy reduces cellular injury by blocking the induction of apoptosis and inhibiting the activation of apoptosis-related caspase [42]. Wang et al. showed that autophagy mediates the initiation of apoptosis and protects cells from necrosis in low-concentration BEAS-2B cells exposed to crotonaldehyde [43]. Xi et al. found that the autophagy signalling pathway could regulate apoptosis. The basic physiological functions of cells are maintained, and the damage to the body caused by stress is reduced when the interaction between autophagy and apoptosis reaches a dynamic balance [44]. However, excessive autophagy could contribute to the induction of apoptosis, which could aggravate cell damage and death under hypoxia conditions.

### 3.3. PRL Regulated Oxidative Stress, Autophagy, and Apoptosis by Binding with L-PRLR and S-PRLR

PRL exerts its action in lactation, immunity, and reproductive regulation [45,46] by first binding to the target cells receptors situated on the plasma membrane. The prolactin receptor *PRLR* plays a central role in the PRL signal transduction cascade, as PRL exerts its biological functions by binding to *PRLR* [47]. *PRLR* belongs to the class 1 cytokine receptor superfamily, characterized by its possession of an extracellular ligand-binding domain, a single pass transmembrane chain, and an intracellular domain [48]. Ruminants have been reported to have long and short prolactin receptors (*L-PRLR*, *S-PRLR*, respectively) [47]. The expression of *L-PRLR* and *S-PRLR* is achieved through alternative splicing of a single PRLR gene. *S-PRLR* differs from *L-PRLR* in that it has a 39-base pair inserted at the beginning of the cytoplasmic domain and two consecutive intra-frame terminating codons at its 3′ end [15]. Oxidative stress in offspring mice was found to be associated with an increased expression of *PRLR* in the hypothalamus and adipose tissue due to the restrictive or hypercaloric diets of maternal mice [49]. Dandawate et al. found that penfluridol could block the expression of *PRLR* in pancreatic cancer cells leads to a reduction in cell proliferation, induction of autophagy, and deceleration of xenograft tumour growth in mice [50]. Tumour-related cell death can be regulated through autophagy [51] and apoptosis [52] by targeting PRL/PRLR, while the combination of PRL and *PRLR* has a dose–response relationship: a low concentration of PRL promotes the secretion of *PRLR*, whereas a high dose of PRL inhibits *PRLR* [53]. This is consistent with the results of this study. In this study, the expression levels of *L-PRLR* and *S-PRLR* decreased after the addition of 500 ng/mL PRL. More importantly, the ROS content, cell apoptosis, and the expression of the autophagy-related proteins (ATG5 and ATG7) increased, while the content of MRCC I–V decreased after the knockdown of *L-PRLR* and *S-PRLR* in high PRL concentration GCs, respectively. However, the results were opposite after overexpression of *L-PRLR* and *S-PRLR* in high PRL concentration GCs, indicating that high PRL concentration promotes oxidative stress, autophagy, and apoptosis by downregulating *L-PRLR* and *S-PRLR*.

Infertility is a major cause of economic losses and a major limitation to achievement of optimum efficiency in the livestock production system [54]. Meanwhile, most infertile sheep are accompanied by significantly elevated levels of PRL in the blood [55]. In humans, hyperprolactinemia is one of the most common causes of infertility in women [56]. In the present study, we first identified that high concentrations of PRL lead to apoptosis through oxidative stress and autophagy, and subsequent studies on *PRLR*s showed that the oxidative stress and apoptosis significantly decreased to a normal level after overexpression of *PRLR*s, while the result was the opposite after interference of *PRLR*s, suggesting that overexpression of *PRLR*s could reduce the level of oxidative stress and apoptosis in high PRL concentration GCs. These findings provide a theoretical basis for the mechanism by which high PRL concentration regulates ovarian GCs oxidative stress and apoptosis, providing a basis for future gene editing interventions in addressing cases of reproductive performance resulting from ovulation failure in sheep caused by oxidative stress and high levels of PRL.

## 4. Materials and Methods

### 4.1. Experimental Design

The high-concentration PRL ovine ovarian GCs model was established, and antioxidant glutathione was used to explore whether high PRL concentration caused apoptosis through oxidative stress by detecting the relevant indicators. PRL plays the important role in ovine ovary through combining with *L-PRLR* and *S-PRLR*; however, the effects of *L-PRLR* and *S-PRLR* in ovaries are different. We used gene editing technology to knock down and overexpress *L-PRLR* and *S-PRLR*, respectively, to explore whether high PRL concentration regulates oxidative stress through *L-PRLR* and *S-PRLR*.

### 4.2. Experiment 1: The Optimal Concentration of Glutathione to Promote GC Proliferation

#### 4.2.1. Cell Proliferation and Expression of Apoptosis-Related Genes

GCs frozen in liquid nitrogen were recovered and seeded in 96-well culture plates at a density of 1 × 10^4^ per well and treated with 0, 1, 2, 5, or 10 μmol/mL reduced glutathione for 24 h, respectively. Cells were treated with 10 μL CCK-8 and incubated in darkness for 1–4 h. Subsequently, the optical density at 450 nm was measured using a microplate reader to determine the optimum glutathione concentration. The optimal glutathione concentration was 5 μmol/mL.

GCs were seeded with a density of 1 × 10^5^ per well into six-well plates and incubated with 0 (C group) and 5 (T group) μmol/mL glutathione for 24 h. The GCs in the C and T groups were trypsinised and collected in 1.5 mL centrifuge tubes for the expression detection of apoptosis-related genes (*Bcl-2*, *Bax* and *caspase 3*).

#### 4.2.2. RNA Extraction and Quantitative Real-Time Polymerase Chain Reaction (PCR)

Primer Premier 5.0 software was used to custom-design the primers of *Bcl-2*, *Bax*, *caspase 3*, *L-PRLR*, *S-PRLR,* and *GAPDH*, based on the conserved region. The synthesis of these primers was entrusted to Shanghai Sheng Gong Biotechnology Co., Ltd. (Shanghai, China). The corresponding information is provided in Appendix A. Total RNA was isolated using Trizol reagent (Invitrogen, Carlsbad, CA, USA) and subjected to reverse transcription using a reagent kit (Takara, 6215A, Beijing, China), following the manufacturer’s protocols. The 5× Mix (4 μL), RNA (2 μg), and RNase-free water (16 μL) were combined to form a final volume of 20 µL reaction mixture. The reaction was carried out at 42 °C for 15 min, and then by a quick exposure to 85 °C for 5 s. Subsequently, the product was stored at −20 °C. The quantitative PCR was meticulously performed following the instructions of the LC-480 PCR system, utilizing an Ultra SYBR Mixture (Cowin Biotech Co., Ltd., CW0957M, Taizhou, China). The cycling protocol involved an initial step of 10 s at 95 °C, followed by 40 cycles comprising 15 s at 95 °C and 60 s at 60 °C. The reaction system is shown in Appendix A. The 2^−ΔΔCT^ method was employed to calculate the relative expression of the genes, utilizing the results obtained from quantitative real-time PCR.

### 4.3. Experiment 2: The Mechanism of Apoptosis Induced by High PRL Concentration through Oxidative Stress

#### 4.3.1. Sample Collection

To explore whether a high PRL concentration leads to the apoptosis of GCs through oxidative stress and its possible mechanism, 500 ng/mL of PRL (Ovine PRL: PROSPEC, cyt-240, purity ≥ 99.0%) was used as the high PRL concentration for its lower cell viability and higher oxidative stress in ovine ovarian GCs [28,57]. The GCs in the C (0 ng/mL PRL), P (the GCs in C group treated with 500 ng/mL PRL for 24 h), and P-GSH groups (the GCs in P group treated with 5 μmol/mL glutathione for 24 h) were collected with three replicates in each group.

#### 4.3.2. Lentivirus Envelope and GC Infection

*L-PRLR* and *S-PRLR* knockdown lentivirus and knockdown control lentivirus were constructed using lenticrispr vector2 (Appendix A). A pGWLV10-new vector (Appendix A) was utilized to generated lentiviruses for overexpression of *L-PRLR* and *S-PRLR*, as well as control overexpression lentivirus. All synthetic plasmids carrying the lentivirus were obtained from Jiangsu Genewiz Biotechnology Co., Ltd. (Jiangsu, China). For infection, a density of 1 × 10^6^ cells per well was seeded in a six-well plate, followed by infection with the optimal multiplicity of infection (MOI = 400) for each lentivirus group upon reaching 20% confluence. In the knockdown by CRISPR/Cas9, the knockdown sequences (sgRNA) targeting *L-PRLR* and *S-PRLR* were designed (https://benchling.com/signin, accessed on 29 April 2022), as shown in Appendix A. The GCs in P group were incubated with lentivirus for 24 h. Then, to achieve stable infection of GCs, 1.5 μg/mL of puromycin (Sigma, St. Louis, MO, USA) was supplemented in the culture medium for a duration of seven days. In *L-PRLR* knockdown lentivirus-infected GCs of the P group, the *L-PRLR* and *S-PRLR* knockdown lentiviruses, along with the control knockdown lentivirus, were denoted as the P-sg-L, P-sg-SL, and P-sg (negative control) groups, respectively. In overexpression, the overexpressed sequences of *L-PRLR* and *S-PRLR* are shown in Appendix A. The lentivirus was replaced by fresh growth medium after 24 h of incubation. Cellular samples were harvested at 72 h post-infection for assessment of the overexpression result. The GCs of the P group were infected with lentiviruses overexpressing *L-PRLR*, *S-PRLR*, and the control group overexpressing vector, referred to as the P-10-L, P-10-S, and P-10 (negative control group) groups, respectively, with each group comprising three replicates.

The expression of *L-PRLR* and *S-PRLR* was detected using qPCR to evaluate the success of knockdown and overexpression.

#### 4.3.3. Mitochondrial Complex Activity

Approximately 1 mL of mitochondrial extract was introduced to a population of 5 × 10^6^ cells. The sample underwent ice-cold homogenization using either a homogenizer or mortar. Subsequently, the mixture was centrifuged at 4 °C and 600× *g* for 10 min.

Mitochondrial complex I (NADH-Q reductase) (BC0510, Solarbio, Beijing, China), II (FADH2-Q reductase) (BC3230, Solarbio, Beijing, China), III (cytochrome reductase) (BC3240, Solarbio, Beijing, China), IV (cytochrome oxidase) (BC0940, Solarbio, Beijing, China), V (adenosine triphosphate (ATP) synthase) (BC1440, Solarbio, Beijing, China) activities, and ATP content (BC0300, Solarbio, Beijing, China), were measured using their respective analytical kits according to the manufacturer’s protocols [58,59,60].

#### 4.3.4. ROS Detection

The oxidant-sensitive fluorescent dye DCFH-DA-based ROS assay kit was used to determine intracellular ROS levels according to the manufacturer’s instructions. The GCs in the C, P, P-GSH, P-sg-L, P-sg-SL, P-10-L, and P-10-S groups were cultured in 96-well plates and then incubated with DCFH-DA for 30 min in the dark at 37 °C. Post-incubation, the cells were thoroughly washed thrice with DMEM/F12. Subsequently, fluorescence images were acquired using an inverted microscope (Leica DM IRB, Leica, Wetzlar, Germany) equipped with a specific filter, followed by image analysis utilizing ImageJ 1.48v software (National Institutes of Health, Bethesda, MD, USA, http://imagej.nih.gov, accessed on 10 November 2022).

#### 4.3.5. Oxidative Stress Assay

The GCs in the C, P, P-GSH, P-sg-L, P-sg-SL, P-10-L and P-10-S groups were trypsinized and collected in 1.5 mL centrifuge tubes. After collection, the cells were immediately centrifuged twice at 1500× *g* for 10 min to remove the supernatant and obtain the GCs, which were kept at −80 °C prior to analysis.

Superoxide dismutase (SOD) activity (BC0170, Solarbio, Beijing, China), total antioxidant capacity (T-AOC) (BC130, Solarbio, Beijing, China), and thioredoxin peroxidase (TPx) (G0217W, Geruisi, Suzhou, China) activities were measured following the manufacturer’s instructions using their respective analytical kits.

#### 4.3.6. Apoptosis and Cell Activity Assay

The GCs in the C, P, P-GSH, P-sg-L, P-sg-SL, P-10-L, and P-10-S groups were inoculated into 6-well plates at a density of 1 × 10^5^ per well for 24 h, respectively. Each condition was subjected to three independent biological replicates. Following cell collection, a single rinse with pre-cooled phosphate-buffered saline (PBS) was performed. The cells were then resuspended in a pre-cooled binding buffer, followed by the addition of Annexin V-FITC. The cell suspension was gently mixed and incubated at room temperature for 15 min. The cells were harvested by centrifugation at 1500 rpm for 5 min. After removing the supernatant, the cells were resuspended in a pre-cooled binding buffer. Subsequently, PI staining solution was added and gently mixed with the cell suspension, followed by storage at 4 °C in the dark. Apoptosis was quantified within 1 h using a BD FACSCanto II flow cytometer.

GCs in the C, P, P-GSH, P-sg-L, P-sg-SL, P-10-L and P-10-S groups were seeded in 96-well-culture plates at a density of 1 × 10^4^ per well. Then, a 10 μL CCK-8 was added into each well, and the cells were hatched in the dark for 1–4 h. A microplate reader was used to quantify the optical density of the cells at 450 nm. Cell activity values were calculated based on test data. Cell survival rate γ calculation common formula is γ = [(As − Ab)/(Ac − Ab)] × 100% (As: optical density of the P group, Ac: optical density of the C group, Ab: optical density of the blank well).

#### 4.3.7. Western Blotting

The total proteins from the C, P, P-GSH, P-sg-L, P-sg-SL, P-10-L, and P-10-S groups were lysed for 30 min using protease inhibitor (PMSF). Subsequently, the lysates were subjected to 10% sodium dodecyl sulphate-polyacrylamide gel electrophoresis (SDS-PAGE) for separation. The membranes underwent a standard blocking step using 5% nonfat milk, hybridised with primary antibody (ATG5, 1:500, R23497; ATG7, 1:500, R23498; GAPDH, 1:5000, 200306-7E4) at 4 °C overnight. Following three washes with PBS, the membranes were incubated with a secondary antibody (goat anti-rabbit IgG, diluted at 1:500) for 1 h at room temperature. The membranes were then processed using an enhanced chemiluminescence detection system (Beyotime Biotech, Shanghai, China) and assessed utilizing a gel imaging system (C500, Azure Biosystems, Dublin, CA, USA). ImageJ software (https://imagej.nih.gov/ij/, accessed on 10 November 2022) was used for grayscale analysis.

### 4.4. Statistical Analysis

Statistical analysis was conducted using SPSS software 22.0. The data were presented as the least squares means and standard errors. The results obtained using a reverse transcription PCR (RT-PCR) study were statistically analysed using Student’s *t*-test between groups C and T. The cell proliferation and apoptosis, the expression of genes and proteins, and other relevant indicators among multiple groups were analysed through the one-way analysis of variance, and the statistical differences were assessed by Duncan’s test. *p* values were adjusted for multiple comparisons across various groups using the Bonferroni correction method. Statistical significance was considered at a threshold of *p* < 0.05. GraphPad Prism 9.0 software was employed for data visualization and mapping.

## 5. Conclusions

In summary, high PRL concentrations downregulated the expression of *L-PRLR* and *S-PRLR*, further inhibiting the activity of MRCC I–V and enhancing ROS positive rate and the expression of ATG5 and ATG7 in ovine ovarian GCs. Conversely, upregulation of *L-PRLR* and *S-PRLR* suppressed oxidative stress and cellular autophagy, thereby alleviating cell apoptosis.

## Figures and Tables

**Figure 1 ijms-24-14407-f001:**
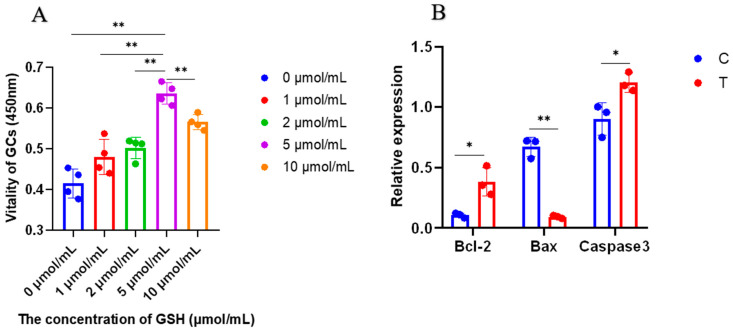
Proliferation and apoptosis of GCs. “*” and “**” indicate 0.01 < *p* < 0.05 and *p* < 0.01, respectively. (**A**) GCs was seeded in 96-well culture plates at a density of 1 × 10^4^ per well, with four replicates in each group; (**B**) GCs was seeded in 6-well culture plates at a density of 1 × 10^5^ per well, with three replicates in each group. The vitality of GCs were analysed through the one-way analysis of variance, and the statistical differences were assessed by Duncan’s test. The 2^−ΔΔCT^ method was employed to calculate the relative expression of the genes, utilizing the results obtained from quantitative real-time PCR. The results were statistically analysed using Student’s *t*-test between groups C and T. C group: Control group; T group: GCs with 5 μmol/mL glutathione. GCs: Granulosa cells.

**Figure 2 ijms-24-14407-f002:**
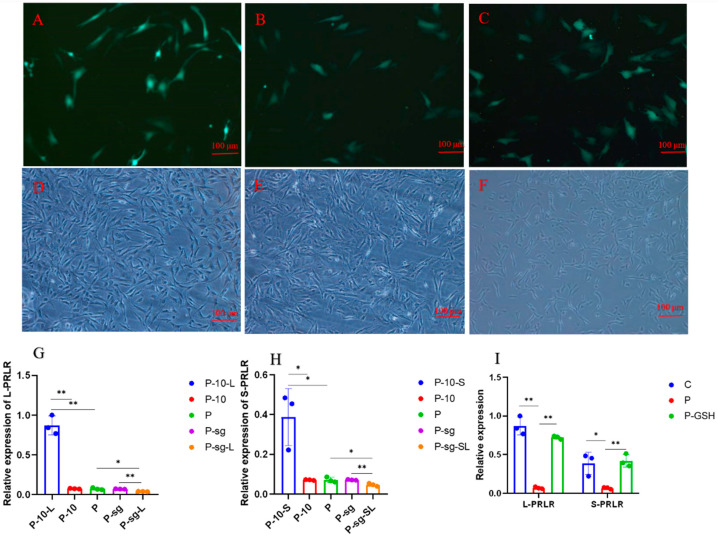
Identification of knockdown and overexpression of GCs. “*” and “**” indicate 0.01 < *p* < 0.05 and *p* < 0.01, respectively. (**A**) Fluorescence of P-10 group; (**B**) fluorescence of P-10-L group; (**C**) fluorescence of P-10-S group; (**D**) positive cells in P-sg group; (**E**) positive cells in P-sg-L group; (**F**) positive cells in P-sg-SL group; (**G**) The relative expression of *L-PRLR* after knocked down and overexpressed of *L-PRLR*; (**H**): The relative expression of S*-PRLR* after knocked down and overexpressed of S*-PRLR*; (**I**): The relative expression of *L-PRLR* and *S-PRLR*. GCs was seeded in 6-well culture plates at a density of 1 × 10^5^ per well, with three replicates in each group. The 2^−ΔΔCT^ method was employed to calculate the relative expression of the genes, utilizing the results obtained from quantitative real-time PCR. The expression of genes were analysed through the one-way analysis of variance, and the statistical differences were assessed by Duncan’s test. C group: Control group; P group: GCs treated with 500 ng/mL PRL (GCs with high PRL concentration); P-GSH group: GCs with high PRL concentration treated with 5 µmol/mL glutathione; P-sg: GCs with high PRL concentration were infected by empty plasmid (negative control group for knockdown); P-sg-L: GCs with high PRL concentration were infected by lentiviruses carrying knockdown sequences of *L-PRLR*; P-sg-SL: GCs with high PRL concentration were infected by lentiviruses of knock down both *L-PRLR* and *S-PRLR*; P-10: GCs with high PRL concentration were infected by empty vectors of overexpression (negative control group for overexpression); P-10-L: GCs with high PRL concentration were infected by lentiviruses carrying overexpressed sequences of *L-PRLR*; P-10-S: GCs with high PRL concentration were infected by lentiviruses carrying overexpressed sequences of *S-PRLR*. GCs: Granulosa cells.

**Figure 3 ijms-24-14407-f003:**
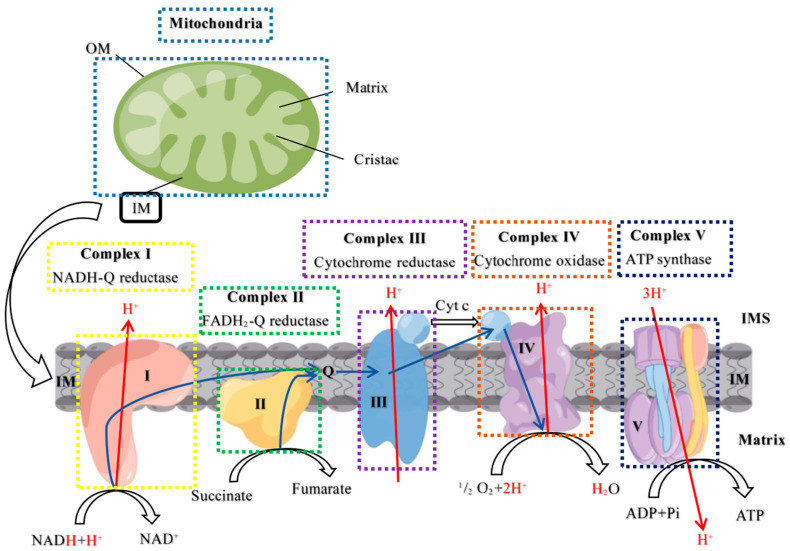
Mitochondrial electron transport chain. OM: Outer membrane; IM: intermembrane; IMS: intermembrane space; Cyt c: Cytochrome C; Q: Ubiquinone Q.

**Figure 4 ijms-24-14407-f004:**
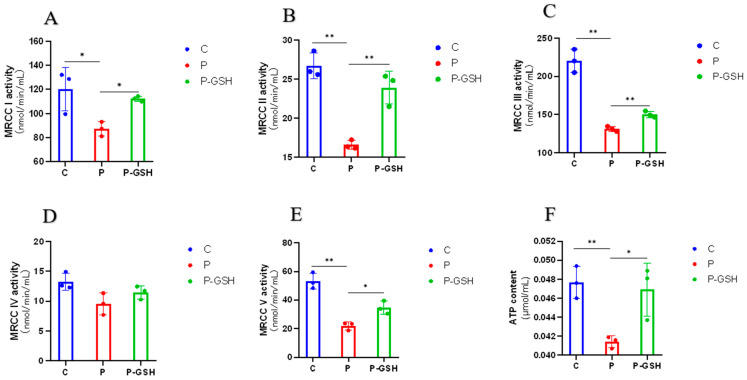
The activity of mitochondrial complex. “*” and “**” indicate 0.01 < *p* < 0.05 and *p* < 0.01, respectively. (**A**–**E**) The activity of MRCC I–V; (**F**) The content of ATP. C group: Control group; P group: GCs treated with 500 ng/mL PRL (GCs with high PRL concentration); P-glutathione group: GCs with high PRL concentration treated with 5 µmol/mL glutathione. GCs was seeded in 6-well culture plates at a density of 1 × 10^5^ per well, with three replicates in each group. The activity of MRCCI–V and the content of ATP were analysed through the one-way analysis of variance, and the statistical differences were assessed by Duncan’s test. MRCC I: Mitochondrial complex I (NADH-Q reductase); MRCC II: Mitochondrial complex II (FADH2-Q reductase); MRCC III: Mitochondrial complex (Cytochrome reductase); MRCC IV: Mitochondrial complex (Cytochrome oxidase); MRCC V: Mitochondrial complex (ATP synthase).

**Figure 5 ijms-24-14407-f005:**
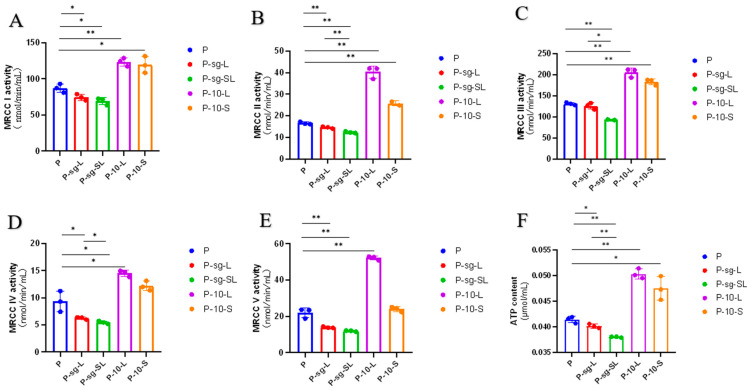
The activity of mitochondrial complex. “*” and “**” indicate 0.01 < *p* < 0.05 and *p* < 0.01, respectively. (**A**–**E**) The activity of MRCC I–V; (**F**) The content of ATP. P group: GCs treated with 500 ng/mL PRL (GCs with high PRL concentration); P-sg-L group: GCs with high PRL concentration were infected by lentiviruses carrying knockdown sequences of *L-PRLR*; P-sg-SL group: GCs with high PRL concentration were infected by lentiviruses of knockdown in both *L-PRLR* and *S-PRLR*; P-10-L group: GCs with high PRL concentration were infected by lentiviruses carrying overexpressed sequences of *L-PRLR*; P-10-S group: GCs with high PRL concentration were infected by lentiviruses carrying overexpressed sequences of *S-PRLR*. GCs was seeded in 6-well culture plates at a density of 1 × 10^5^ per well, with three replicates in each group. The activity of MRCCI–V and the content of ATP were analysed through the one-way analysis of variance, and the statistical differences were assessed by Duncan’s test. MRCC I: Mitochondrial complex I (NADH-Q reductase); MRCC II: Mitochondrial complex II (FADH2-Q reductase); MRCC III: Mitochondrial complex (Cytochrome reductase); MRCC IV: Mitochondrial complex (Cytochrome oxidase); MRCC V: Mitochondrial complex (ATP synthase).

**Figure 6 ijms-24-14407-f006:**
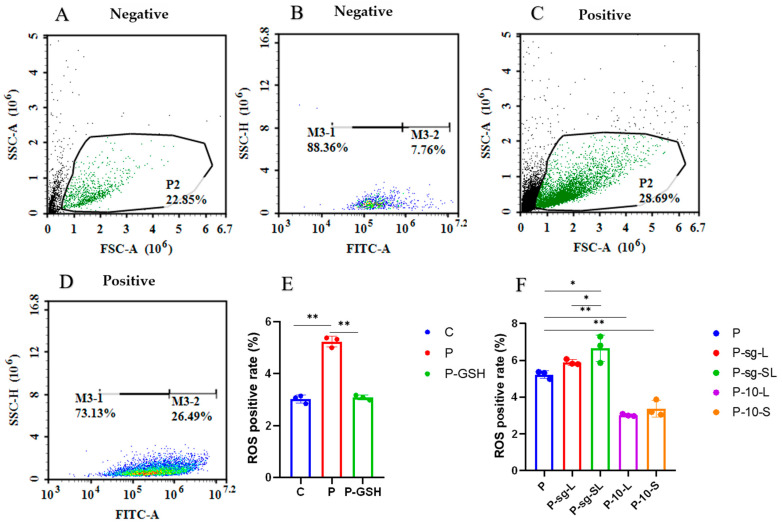
The detection of ROS. “*” and “**” indicate 0.01 < *p* < 0.05 and *p* < 0.01, respectively. (**A**) Negative control of FSC-A; (**B**) negative control of FITC-A; (**C**) positive control of FSC-A. (**D**) positive control of FITC-A; (**E**,**F**) ROS positive rate (%). GCs was seeded in 96-well culture plates at a density of 1 × 10^4^ per well, with three replicates in each group. The relative fluorescence of ROSs were analysed through the one-way analysis of variance, and the statistical differences were assessed by Duncan’s test. C group: Control group; P group: GCs treated with 500 ng/mL PRL (GCs with high PRL concentration); P-GSH group: GCs with high PRL concentration treated with 5 µmol/mL glutathione; P-sg-L group: GCs with high PRL concentration were infected by lentiviruses carrying knockdown sequences of *L-PRLR*; P-sg-SL group: GCs with high PRL concentration were infected by lentiviruses of knockdown in both *L-PRLR* and *S-PRLR*; P-10-L group: GCs with high PRL concentration were infected by lentiviruses carrying overexpressed sequences of *L-PRLR*; P-10-S group: GCs with high PRL concentration were infected by lentiviruses carrying overexpressed sequences of *S-PRLR*. GCs: Granulosa cells; ROS: the reactive oxygen species.

**Figure 7 ijms-24-14407-f007:**
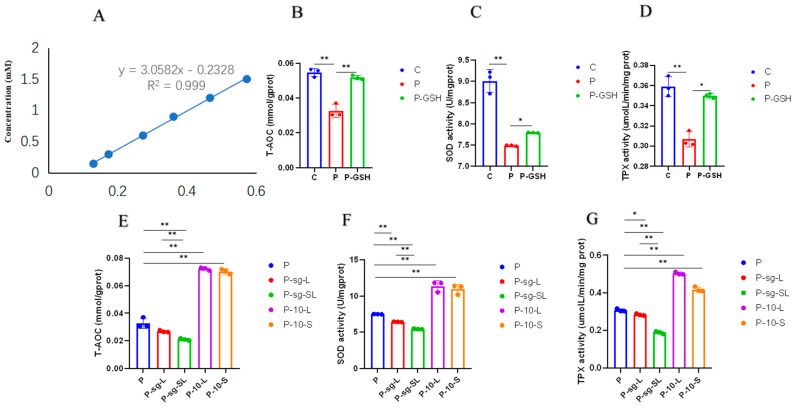
Oxidative stress parameters. “*” and “**” indicate 0.01 < *p* < 0.05 and *p* < 0.01, respectively. (**A**) The standard curve of TAC; (**B**): T-AOC in C, P and P-GSH groups; (**C**): The activity of SOD in C, P and P-GSH groups; (**D**): The activity of TPX in C, P and P-GSH groups; (**E**): T-AOC in P, P-sg-L, P-sg-SL, P-10-L and P-10-S groups; (**F**): The activity of SOD in P, P-sg-L, P-sg-SL, P-10-L and P-10-S groups; (**G**): The activity of TPX in P, P-sg-L, P-sg-SL, P-10-L and P-10-S groups. GCs was seeded in 6-well culture plates at a density of 1 × 10^5^ per well, with three replicates in each group. SOD activity, T-AOC, and TPx activity were analysed through the one-way analysis of variance, and the statistical differences were assessed by Duncan’s test. C group: Control group; P group: GCs treated with 500 ng/mL PRL (GCs with high PRL concentration); P-GSH group: GCs with high PRL concentration treated with 5 µmol/mL glutathione; P-sg-L group: GCs with high PRL concentration were infected by lentiviruses carrying knockdown sequences of *L-PRLR*; P-sg-SL group: GCs with high PRL concentration were infected by lentiviruses of knockdown in both *L-PRLR* and *S-PRLR*; P-10-L group: GCs with high PRL concentration were infected by lentiviruses carrying overexpressed sequences of *L-PRLR*; P-10-S group: GCs with high PRL concentration were infected by lentiviruses carrying overexpressed sequences of *S-PRLR*. T-AOC: total antioxidant capacity; SOD: Superoxide dismutase; TPx: Thioredoxin peroxidase; GCs: Granulosa cells.

**Figure 8 ijms-24-14407-f008:**
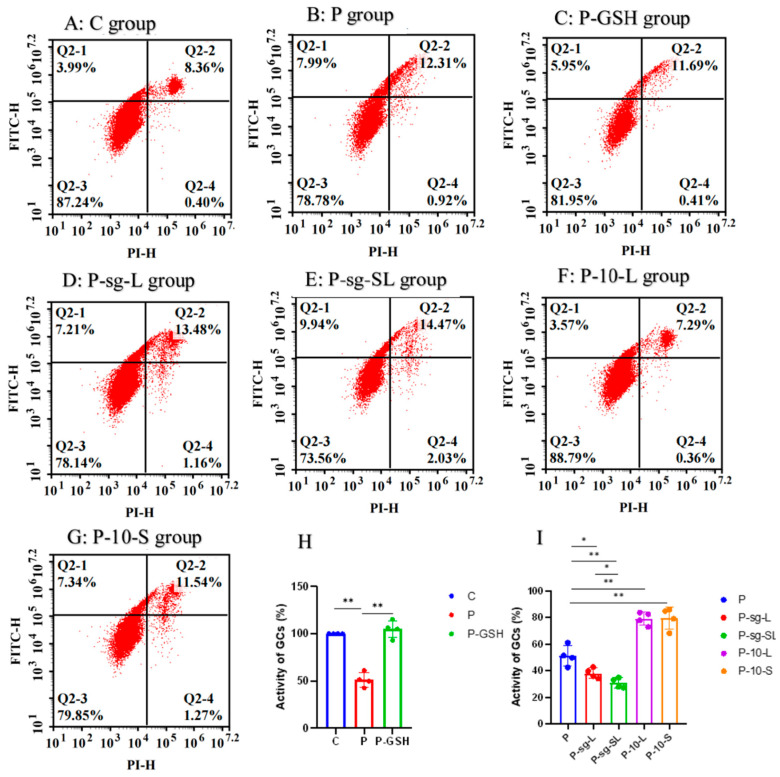
Apoptosis and cell activity in cultured ovine ovarian granulosa cells. “*” and “**” indicate 0.01 < *p* < 0.05 and *p* < 0.01, respectively. (**A**–**G**) The X axis represents PI fluorescence; Y axis represents Annexin-V fluorescence. Q2-1: the dead cells; Q2-2: the early withered; Q2-3: the living cells; Q2-4: the late withered; (**H**): Cell activity of GCs in C, P and P-GSH groups; (**I**): Cell activity of GCs in P, P-sg-L, P-sg-SL, P-10-L and P-10-S groups. (**A**–**G**): GCs was seeded in 6-well culture plates at a density of 1 × 10^5^ per well, with four replicates in each group; (**H**) GCs was seeded in 96-well culture plates at a density of 1 × 10^4^ per well, with four replicates in each group. The activity of GCs were analysed through the one-way analysis of variance, and the statistical differences were assessed by Duncan’s test. C group: Control group; P group: GCs treated with 500 ng/mL PRL (GCs with high PRL concentration); P-GSH group: GCs with high PRL concentration treated with 5 µmol/mL glutathione; P-sg-L group: GCs with high PRL concentration were infected by lentiviruses carrying knockdown sequences of *L-PRLR*; P-sg-SL group: GCs with high PRL concentration were infected by lentiviruses of knockdown in both *L-PRLR* and *S-PRLR*; P-10-L group: GCs with high PRL concentration were infected by lentiviruses carrying overexpressed sequences of *L-PRLR*; P-10-S group: GCs with high PRL concentration were infected by lentiviruses carrying overexpressed sequences of *S-PRLR*. GCs: Granulosa cells.

**Figure 9 ijms-24-14407-f009:**
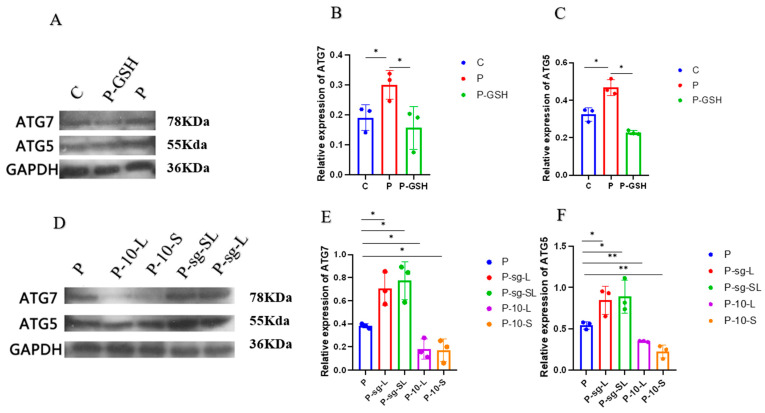
Detection of autophagy-associated proteins. “*” and “**” indicate 0.01 < *p* < 0.05. (**A**,**D**): The results of WB; (**B**): The relative expression of ATG7 in C, P, P-GSH groups; (**C**): The relative expression of ATG5 in C, P, P-GSH groups; (**E**): The relative expression of ATG7 in P, P-sg-L, P-sg-SL, P-10-L and P-10-S groups; (**F**): The relative expression of ATG5 in P, P-sg-L, P-sg-SL, P-10-L and P-10-S groups. GCs was seeded in 6-well culture plates at a density of 1 × 10^5^ per well, with three replicates in each group. SOD activity, T-AOC, and TPx activity were analysed through the one-way analysis of variance, and the statistical differences were assessed by Duncan’s test. C group: Control group; P group: GCs treated with 500 ng/mL PRL (GCs with high PRL concentration); P-GSH group: GCs with high PRL concentration treated with 5 µmol/mL glutathione; P-sg-L group: GCs with high PRL concentration were infected by lentiviruses carrying knockdown sequences of *L-PRLR*; P-sg-SL group: GCs with high PRL concentration were infected by lentiviruses of knockdown in both *L-PRLR* and *S-PRLR*; P-10-L group: GCs with high PRL concentration were infected by lentiviruses carrying overexpressed sequences of *L-PRLR*; P-10-S group: GCs with high PRL concentration were infected by lentiviruses carrying overexpressed sequences of *S-PRLR*. T-AOC: total antioxidant capacity; SOD: Superoxide dismutase; TPX: Thioredoxin peroxidase; GCs: Granulosa cell.

**Table 1 ijms-24-14407-t001:** The apoptosis rate of ovine ovarian GCs (%).

Items	Group							SEM	*p*-Value
	C	P	P-GSH	P-sg-L	P-sg-SL	P-10-L	P-10-S		
Early apoptotic cells rate	7.55 ^d^	12.57 ^b^	11.45 ^c^	14.31 ^a^	14.71 ^a^	7.25 ^d^	10.55 ^c^	0.366	<0.001
Late apoptotic or necrotic cells rate	0.35 ^d^	0.97 ^c^	0.39 ^d^	1.17 ^b^	2.08 ^a^	0.38 ^d^	1.16 ^b^	0.0342	<0.001
Total apoptotic rate	7.91 ^e^	13.87 ^c^	11.84 ^d^	15.48 ^b^	16.79 ^a^	7.67 ^e^	11.71 ^d^	0.386	<0.001

SEM: standard error of the mean. The different lowercase letters indicate significant differences (*p* < 0.05).

## Data Availability

All datasets generated during and analysed during the current study are available from the corresponding author (zhangyingjie66@126.com) on reasonable request.

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
