# Peer review of "High Prolactin Concentration Induces Ovarian Granulosa Cell Oxidative Stress, Leading to Apoptosis Mediated by L-PRLR and S-PRLR"

_ijms, 2023, doi:10.3390/ijms241914407_

Round 1

Reviewer 1 Report

Young and colleagues found and discuss the possibility that high concentration of prolactin induces apoptotic cell death in ovine ovarian GCs by downregulating L-PRLR and S-PRLR, activating oxidative stress and autophagic pathways.

This manuscript is very interesting for this reviewer however, the authors (almost same authors) recently published as below:

Biochemistry, Biophysics and Molecular Biology

Effect of prolactin on cytotoxicity and oxidative stress in ovine ovarian granulosa cells

Ruochen Yang1, Shuo Zhang2, Chunhui Duan1, Yunxia Guo1, Xinyu Shan1, Xinyan Zhang1, Sicong Yue1, Yingjie Zhang1, Yueqin Liu1

Published July 10, 2023

https://peerj.com/articles/15629/

(Conclusion)

A low concentration of PRL inhibited cytotoxicity and oxidative stress. HPC induced oxidative stress in ovine ovarian GCs via the pentose phosphate pathway by modulating the associated proteins MT-ND1 in ROS pathway and UBA52, MAPK12 and BCL2L1 in mitophagy pathway, resulting in cytotoxicity.

For this reviewer, the contents of this journal and recently published manuscript are very similar, and some results are overlapped. Furthermore, there are no explanation about the relation between the date the authors recently published and the data described in this submitting manuscript, and no discussion. The authors should include these description in this manuscript. This reviewer recommends editors to ask authors to make major revise and resubmit as a new submission.

Before accepting this manuscript, this reviewer needs to review again.  

This reviewer did not check for English errors, please ask somebody professional to check English used in this entire document, figures and figure legends before submission. 

Comments:

Page 2, line 80-81: …..first, then it decreased (Figure 1A)

What does it mean "first"? The graph showed the relation between the concentration of GSH and cell viability. How long did authors treat the cells with GSH? Did authors observe "time course effect of GSH on the cells"?

Page 3, line 88: … C and T….

What do they mean "C and T" groups? Unfortunately, the section of "Results" and " Materials and Methods" located far away, for readers, it is hard to go back and forth to make sure what "C" and "T" represents. The authors should describe what these abbreviations mean in the figure 1 legend, otherwise for readers it is very hard to understand what each represents.

Page 3, line 93: Cell fluorescence and the expression of L-PRLR and S-PRLR

It is very hard for this reviewer to understand what the results the authors want to show here from the section title. As this reviewer indicated that unfortunately, most of reader will not read materials and method first, because of the description order of "Result" and "Materials and Methods". Authors should pay attention more carefully, if authors want reader to be more comfortable and get their attentions. Please check all section title, This reviewer feels same as written here.

Line 103: ….. after the knockdown of L-PRLR ……..

Why did author need to knock down?

The authors need some description of why the authors did this step, and what the purpose the authors want to show, otherwise it is very difficult to follow the logic that authors took. In the manuscript, the authors need transition sentences to make the flow-throw of document flow better. This reviewer recommends working with professional English editors before submitting document.

Page 4, line 184: ….. with 500ng/mL…

Is that effected by dose-dependent manner? How did authors select this dose? There is no description in Materials methods and in the document.

Page 11, line 273: .. a Takara kit….

“Takara” is the company name, not kit’s name. The authors must write correct kit’s name and reference following journal rule.

??????? Kit (Company, location)

Line 278: Ultra SYBR Mixture

The authors must add reference following journal rule if it was purchased from industry.

Line 285: 500 ng/mL PRL

Why did authors use this dose?

Page 12, line 324: the activities of…..

“the” should be “The”

Reviewer 2 Report

In this work, the Authors aimed to examine the mechanism of apoptosis induced by HPC in GCs. They found that HPC induces apoptotic cell death in ovine ovarian GCs by downregulating L-PRLR and S-PRLR, activating oxidative stress and autophagic pathways. The work sounds scientifically original. The discussion of the results is convincing. There are some discrepancies and minor corrections that need to be addressed.

Lines 62-63: Since autophagy and apoptosis are essential points of the paper, I suggest adding more studies related to these phenomena in proliferation and ovulation in GCs.

Lines 182-184: Explain the meaning of this result in this paragraph.

Lines 251-253: A better explanation of this data is required.  In which way PRL regulates oxidative stress, autophagy, and apoptosis by binding with L-PRLR and S-PRLR?

Line 254: In materials and methods the experimental design is not clear. I suggest better organizing the materials and methods section. Furthermore, references are required in this section.

Lines 365: The conclusions section needs to be improved. 

Reviewer 3 Report

The authors aimed to study apoptosis in the ovary of sheep as the result of oxidative stress.

Overall, the study is useful, so it can advance to the next stage of evaluation.

Major issues

-At the start of the M&M section, please add a new sub-section to describe an overview of the experimental design. As it is the text now, it is confusing and stressing for readers.

-The authors should condense the M&M section by using references to established techniques rather than describing again procedures and methods.

-The Discussion does not touch at all the clinical consequences of the findings. Please add a new sub-section to describe how the findings would affect clinical practice.

Minor issues

-Some of the passages included in Introduction would fit better in Discussion, to explain the findings of the study. However, I leave this to the authors to decide.

-Please make sure that number to literature references in the text correspond correctly with the actual references at the end of the text.

Round 2

Reviewer 1 Report

the authors answered the questions asked by this reviewer and corrected them appropriately.

So this reviewer recommends editors to accept this manuscript for publication however, please check their English before publication.  

Reviewer 3 Report

Point 3: The Discussion does not touch at all the clinical consequences of the findings. Please add a new sub-section to describe how the findings would affect clinical practice. Response 3: Thank you for pointing this out. We have processed the discussion of this study and added the following content: It was found that the oxidative stress and apoptosis significantly decreased to a normal level after overexpression of PRLRs, while the result was the opposite after interference of PRLRs, suggesting that overexpression of PRLRs could reduce the level of oxidative stress and apoptosis in high PRL concentration GCs. These findings provide a theoretical basis for 2 the mechanism by which high PRL concentration regulates ovarian GCs oxidative stress and apoptosis, providing a basis for future gene editing interventions in addressing cases of reproductive performance resulting from ovulation failure in sheep caused by oxidative stress and high levels of PRL.

Can you please extend this passage further and also add some relevant references, please?
Then the manuscript can be published.
